# Effect of Small Volume Blood Sampling on the Outcomes of Very Low Birth Weight Preterm Infants

**DOI:** 10.3390/children9081190

**Published:** 2022-08-08

**Authors:** Pin-Chun Su, Hao-Wei Chung, Shu-Ting Yang, Hsiu-Lin Chen

**Affiliations:** 1Department of Pediatrics, Kaohsiung Medical University Chung-Ho Memorial Hospital, Kaohsiung 80756, Taiwan; 2Department of Respiratory Therapy, College of Medicine, Kaohsiung Medical University, Kaohsiung 80708, Taiwan

**Keywords:** anemia of prematurity, blood transfusion, complication of blood transfusion, phlebotomy

## Abstract

Very low birth weight (VLBW) preterm infants universally experience anemia of prematurity (AOP) while growing up. The effects of reduced blood sample volume on AOP, packed red blood cell (PRBC) transfusion, and outcome in VLBW preterm infants were examined in this study. To reduce blood loss due to phlebotomy, we set up a small volume blood sampling procedure in VLBW infants. In this retrospective study, we compared the VLBW infants who had undergone standard blood sampling (control group, *n* = 20) with those who underwent small volume blood sampling (study group, *n* = 84), with both groups receiving PRBC transfusion under restrictive criteria. Blood loss from phlebotomy and PRBC transfusion volume over 30 days was significantly lower in the study group than in the control group. Compared to the control group, hematocrit, reticulocyte, and iron levels were significantly higher in the study group. There were no significant differences in the proportion of patent ductus arteriosus, severe intraventricular hemorrhage, retinopathy of prematurity, and bronchopulmonary dysplasia between the two groups. The small volume blood sampling resulted in lower PRBC transfusion volume, less severe anemia, and greater bone marrow function at 30 days of age. This strategy can reduce potential adverse effects of PRBC transfusion in VLBW preterm infants.

## 1. Introduction

Anemia of prematurity (AOP) is a common condition in growing preterm infants with very low birth weight (VLBW), and the onset of AOP is inversely proportional to the infant’s gestational age (GA) [1]. The leading cause of AOP is iatrogenic blood loss from phlebotomy; additional factors include impaired erythropoietin (EPO) production, shortened red blood cell life span, and iron deficiency [2]. Many infants with AOP are symptomatic because of a reduced capacity to maintain adequate oxygenation due to poor compensation for AOP. Symptoms associated with AOP include tachycardia, poor weight gain, increased requirements for supplemental oxygen, and an increased apneic episode or bradycardia.

AOP typically resolves within 3–6 months of age; however, some VLBW preterm infants with AOP require interventions such as packed red blood cell (PRBC) transfusion, iron supplementation, administration of erythropoiesis-stimulating agents, and close laboratory monitoring.

PRBC transfusions are primarily used to treat infants with AOP, but potential adverse effects exist, such as blood-borne infectious disease, graft-versus-host disease, metabolic and cardiovascular complications, hypothermia, iron overload, and increased oxidative stress. Increased oxidative stress may be related to the development of complications in preterm infants, such as bronchopulmonary dysplasia (BPD), retinopathy of prematurity (ROP), necrotizing enterocolitis (NEC), intraventricular hemorrhage (IVH), and periventricular leukomalacia (PVL) [3,4,5].

In 2009, we published a study on the effect of blood transfusions on the outcomes of VLBW preterm infants under two different transfusion criteria, namely restrictive and liberal criteria. The restrictive criteria were as follows for the different types of respiratory support that the infants in our study received: to maintain hematocrit >35% in infants with assisted ventilation; hematocrit >30% in infants with non-invasive respiratory support such as nasal continuous positive airway pressure (CPAP) or high flow nasal cannula (HFNC), and hematocrit >22% in infants breathing room air spontaneously. Applying these restrictive criteria to decrease blood transfusions has achieved better clinical outcomes compared to those obtained with liberal criteria. Strategies to decrease the adverse effects of PRBC transfusion include the application of restrictive criteria for PRBC transfusion and reduction of blood loss resulting from phlebotomy [6]. The volume of transfusion was 10 mL/kg each time. The restrictive criteria resulted in a lower PRBC transfusion volume and a higher reticulocyte count at day 30 of life; thus, minimizing the amount of PRBC transfusion may be beneficial in preventing BPD in VLBW infants. Therefore, these restrictive criteria for PRBC transfusions have been used in our policy for VLBW infants in the neonatal intensive care unit (NICU) since 2009.

To further decrease the frequency of PRBC transfusion, in 2016, we adopted small volume blood sampling to reduce blood loss from phlebotomy. Because we have different eras in the blood sampling volume, we would compare the outcomes of very low birth weight preterm infants before and after small volume blood sampling. Hence, this study aimed to examine the effect of blood loss reduction due to phlebotomy on the AOP, PRBC transfusion, and outcomes in VLBW preterm infants.

## 2. Materials and Methods

### 2.1. Study Design and Data Collection

In this retrospective study, preterm infants with GA less than 32 weeks and birth weight less than 1500 g who were admitted to the NICU of Kaohsiung Medical University Hospital (KMUH) between June 2016 and October 2018 were enrolled in the study group who underwent small volume blood sampling. Patients with a chromosomal abnormality or major birth effects were excluded from this study. Infants admitted to our NICU between July 2005 and June 2006, under the restrictive criteria and standard blood sampling in our previous study in 2009 [6], were included in the control group (historical control group). Before 2016, the blood tests for every patient in our hospital should be sent to the central laboratory. At that time, the minimum amount required by the central clinical laboratory at our hospital was as follows: 1 mL for complete blood count (CBC), 1 mL for electrolyte, and 1 mL for blood gas analysis; thus, the maximum of 3 mL for a single blood sample was required for the historical control group in this study. In 2016, we introduced small volume blood sampling to reduce blood loss from phlebotomy by point of care testing. We used an Abbott handheld gas analyzer between June 2016 and January 2017, which was replaced with an ABL80 FLEX blood gas analyzer (Radiometer, Copenhagen, Denmark) in January 2017. Using a small blood volume (0.2 mL), which was 15 times lower than the previous blood sample volume of 3 mL, both gas analyzers revealed data regarding the following parameters: Hct, hemoglobin (Hb), Na+, K+, Ca2+, and gas analysis for infants with small volume blood sampling. For the care of preterm infants, we check CBC, electrolyte, and blood gas analysis when the preterm infants suffer from unstable vital signs or increased apneic episodes and infection or anemia is suspected with the clinical observation. Moreover, we also check electrolytes for monitoring parenteral nutrition and follow up blood gas analysis for respiratory ventilator weaning during the preterm infants’ hospitalization. To evaluate the effects of small volume blood sampling compared with the control group, a CBC was performed on admission and at 30 days of life in this study. In addition, reticulocyte count, lactate, serum iron, and ferritin concentrations were measured at 30 days of life. We also recorded the data for each patient as follows: GA; birth weight; sex; Apgar scores; single or multiple births; daily volume of blood sampled; age at first PRBC transfusion; number and total volume of PRBC transfusions over 30 days; complications; and outcomes. The complications included apnea of prematurity, late-onset sepsis, IVH, ROP, BPD, and NEC. The duration of the hospital stay, the time it took for the infants to regain their birth weight, and the number of days they required respiratory support modalities were all considered when evaluating the outcome.

### 2.2. Ethics Statement

The study was approved by the institutional review board-I of Kaohsiung Medical University Hospital (approval number: KMUHIRB-SV(I)-20180037, date: 13 July 2018). Written informed consent was obtained from the parents of the patients before their discharge from the Kaohsiung Medical University Hospital or at our outpatient clinics.

### 2.3. Statistical Analysis

Mean ± standard deviation was used to express the descriptive results of continuous variables. The two groups were compared using the Wilcoxon test for numerical data and the chi-squared test for categorical data. Multiple logistic regression analysis was performed to investigate the association between total blood sample volume and the clinical outcomes in both groups. *p*-values < 0.05 were considered statistically significant. Statistical analysis was performed using JMP 14.2.0 desktop statistical discovery software (SAS Institute Inc., Oklahoma City, OK, USA).

## 3. Results

### 3.1. Patient Characteristics

Between June 2016 and October 2018, 86 VLBW preterm infants were admitted to our NICU. After the exclusion of one infant due to standard blood sampling and one infant due to death occurring before 30 days of age, 84 VLBW preterm infants were enrolled in the study group. All infants in the study group were under the restrictive criteria for PRBC transfusion. Delayed cord clamping was not performed in either the control group or the study group because of the unstable condition of these preterm infants immediately after birth.

There were no significant differences between the two groups concerning GA, birth weight, the proportion of small for gestational age, sex, multiple births, Apgar scores, and initial Hb level (Table 1).

### 3.2. Packed Red Blood Cell Transfusion, Blood Loss, and Iron Status at Day 30

Infants with small volume blood sampling had significantly lower blood loss volume via phlebotomy and lower PRBC transfusion volume at 30 days of age than infants with standard blood sampling (Table 2). The study group had significantly higher hematocrit levels, reticulocyte, and iron levels than the control group (Table 2). Serum iron levels may be associated with GA, PRBC transfusion volume, feeding volume, and iron supplementation. Serum ferritin levels were lower in the study group than in the control group, but the difference was not significant. Higher lactate levels were observed in the study group than in the control group, although these were within the normal range.

Compared to the control group, a more aggressive feeding policy was applied to the study group. Infants in the study group received a significantly larger total milk volume than those in the control group by day 30 (Table 3).

Multiple regression analysis revealed that feeding volume significantly affected serum iron levels (*p* = 0.002) (Table 4). GA, birth weight, PRBC transfusion volume, and blood loss volume via phlebotomy by day 30 were not associated with serum iron levels at day 30 in this study.

### 3.3. Clinical Outcomes

We further analyzed some common complications of prematurity that may be related to oxidative stress attributed to blood transfusion (Table 5).

There were no significant differences in the proportion of infants with respiratory distress syndrome (RDS) with the need for surfactant therapy, patent ductus arteriosus, severe IVH, ROP, and BPD between the two groups (Table 5). A similar incidence of all stages of necrotizing enterocolitis (NEC) was observed in the study group and the control group. The incidence of confirmed NEC (≥2a Bell’s stage) was lower in the study group than in the control group, but the difference was not significant. The incidence of culture-proven sepsis was significantly lower in the study group than in the control group.

The time to regain birth weight was significantly shorter in the study group. There were no significant differences in the duration of ventilator/nasal CPAP and supplemental oxygen use between the two groups.

### 3.4. Factors for Complications Related to Oxidative Stress

Multiple logistic regression analysis was applied to determine the effects of GA, birth weight, PRBC volume transfused by day 30, and blood loss via phlebotomy by day 30 on the development of IVH, NEC, BPD, and ROP. After controlling for these variables, multiple regression analysis revealed an increase in NEC incidence with increased PRBC transfusion volume over 30 days (*p* = 0.018). PRBC transfusion volume was not significantly associated with other complications (Table 6).

## 4. Discussion

Our study revealed that small volume sampling was associated with a lower PRBC transfusion volume, less severe anemia, and greater bone marrow function for red blood cell production at 30 days of age in VLBW preterm infants. Reducing blood loss from phlebotomy through small volume blood sampling would reduce the potential adverse effects of PRBC transfusion in VLBW preterm infants.

Growing preterm infants frequently develop AOP, primarily caused by iatrogenic blood loss due to phlebotomy for blood testing [7]. PRBC transfusions are the primary intervention used to treat infants with AOP; more than half of preterm infants with a birth weight less than 1250 g receive at least one PRBC transfusion during their hospitalization in the NICU [8]. Although PRBC transfusion promptly and temporarily resolves AOP, there are complications and potential adverse effects, such as metabolic and cardiovascular complications, blood-borne infection, graft-versus-host disease, iron overload, increased oxidative stress, and neurodevelopmental impairment [9]. Complications such as BPD, ROP, NEC, and IVH are associated with increased oxidative stress by PRBC transfusion [3,4,5].

Low blood hematocrit levels in the blood may result in inadequate tissue oxygenation, leading to symptomatic anemia [10] that causes the need for more advanced respiratory support with increased oxygen demands, further increases oxidative stress, which is thought to be the origin of complications related to preterm infants, such as BPD and ROP [3,5,8]. In our study, sampling a small volume of blood resulted in less severe anemia, which could decrease the volume of PRBC transfusions required and further reduce oxidative stress and its associated complications. Although our results showed that VLBW preterm infants in the study group had higher hematocrit and lower volume of PRBC transfusions at 30 days old, there were no significant differences in the rate of BPD and ROP between the two groups. The development of BPD may be due to the excessive oxygen supplementation and the positive pressure with a higher volume delivered by the mechanical ventilator besides oxidative stress from PRBC transfusion [11]. To prevent the development of BPD in VLBW preterm infants, optimizing the mechanical ventilators with supplemental oxygen and also reducing PRBC transfusions are both important [12].

ROP may be influenced by several factors. Shohat et al. first proposed that blood transfusion was a risk factor for ROP [13]; this association was later confirmed by other studies. Recent studies have demonstrated that GA and frequency of blood transfusion are independent risk factors for ROP [14,15,16]. However, in our study, there was no significant difference between the proportion of ROP in the study group and the control group.

NEC in preterm infants may be associated with recent exposure to PRBC transfusion [17], while a higher number and the total volume of PRBC transfusion are associated with an increased risk of confirmed NEC in VLBW infants [18,19,20,21]. Preterm infants who develop transfusion-associated NEC are at a higher risk of mortality, and surgical NEC is more prevalent following PRBC transfusion [19,20,21]. Furthermore, recent data suggest that severe anemia may influence the risk of gut injury [22]; therefore, following care approaches such as delayed cord clamping and reducing blood loss due to phlebotomy are suggested [23]. In our study, the incidence of confirmed NEC was significantly lower in the study group compared to the control group. However, we did not perform delayed cord clamping in both groups. When considering possible risk factors, no association was found between NEC and the following factors: GA, birth weight, sex, Apgar scores at one and five minutes of life, RDS, or PDA. Multiple regression analysis revealed that the risk associated with NEC development in VLBW infants was directly proportional to the overall volume of transfused blood over 30 days (*p* = 0.018) (Table 5), which is consistent with previous reports [18,21].

Plasma non-transferrin bound iron has increased significantly in preterm infants after a blood transfusion. It may also be associated with GA in preterm infants, feeding volume, and iron supplementation [24,25]. Although preterm infants with small volume blood sampling received a significantly lower PRBC transfusion volume at 30 days of age than infants with standard blood sampling in our study, serum iron levels were significantly higher in the study group than in the control group (*p* = 0.003). There were no differences in GA between the two groups, and infants in the study group received a smaller PRBC transfusion volume; therefore, we hypothesized that the higher serum iron levels among infants in the study group were affected by lower blood loss from phlebotomy, and the nutritional support. We adopted a more aggressive feeding policy for infants in the study group due to our advanced knowledge of preterm infant care. As a result, infants in the study group received a significantly larger volume of milk than those in the control group, regardless of the type of milk administered (human milk, donor human milk, or preterm formula) (Table 4). A more aggressive feeding policy also allows the decreased duration of parental nutrition and intravenous access, which may have contributed to the lower incidence of sepsis in the study group. The study group regained the birth weight significantly faster than the control group (*p* = 0.003). Iron supplementation is not used regularly for preterm infants in our NICU, so most of the iron supply comes from human milk or preterm formula. Based on our multiple regression analysis, we postulated that higher serum iron levels among infants in the study group were caused by the increased feeding volume (Table 4). Higher serum iron levels in the study group may be attributed to the advanced nutritional support provided in the NICU. Simultaneously, significantly lower blood loss volume during phlebotomy in infants with small blood volume sampling as compared to infants with previous standard blood volume sampling may increase serum iron level due to the corresponding decrease in iron loss.

In our study, higher lactate levels were found in the study group than in the control group (*p* = 0.043) (Table 2), despite being within the normal range. Possible causes reasons for increased serum lactate levels include a high concentration of amino acids in parenteral nutrition and an aggressive respiratory weaning strategy. A high serum lactate level indicates tissue hypoxia, and an aggressive respiratory weaning strategy may lead to intermittent hypoxia. However, there were no significant differences between the two groups in duration of the ventilator or nasal CPAP use (*p* = 0.073). We further analyzed the potential factors associated with increased lactate levels using multiple regression analysis. This showed that the duration of ventilator/nasal CPAP usage or the duration of supplemental oxygen usage had no effects on serum lactate levels (data not shown).

By using small volume blood sampling with restrictive PRBC transfusion criteria, we increased the quality of care for VLBW preterm infants. The infants in the study group had higher hematocrit, lower volume of PRBC transfusions, and higher reticulocyte levels at 30 days of life. Although there were no differences in the BPD rate and the days of total respiratory support (Table 5), we found that a higher volume of transfused PRBC over 30 days was associated with the development of NEC in VLBW infants.

The strength of this study is that we provide evidence that small blood sampling could lead to a better quality of VLBW preterm care. The limitations of this study were the small sample size and the inclusion of a historical control group (data of premature infants 10 years ago). Due to advances in preterm care, such as a more aggressive feeding policy, a high concentration of amino acids in parenteral nutrition, and an aggressive respiratory weaning strategy, the care for the historical control group was different from the study group. However, conducting a randomized controlled trial in two groups with standard and small blood sampling is unethical because of advances in preterm care. Therefore, we chose the historical control group from the study conducted 10 years ago to complete this study. The concern for including the historical group is about the inequality between these two groups. The condition after birth was determined by the APGAR scores. In this study, we showed no statistical differences in the first minute and fifth minute APGAR scores of the study group and the control group (Table 1). Moreover, no significant differences were observed in the proportion of infants with respiratory distress syndrome (RDS) with the need for surfactant therapy between the two groups (Table 6). Therefore, respiratory condition at birth was comparable to the need for analytical blood determinations. In addition, determining the sample size is challenging because this is a retrospective study. Perhaps due to the sample size, no differences were found in the morbidities of prematurity. We did not exclude those expired infants from this study. However, our institution’s mortality rate has declined compared with 10 years ago; some preterm infants who survived would lead to a higher complication rate such as bronchopulmonary dysplasia than the historical control group.

## 5. Conclusions

Small volume blood sampling resulted in a lower PRBC transfusion volume, less severe anemia, and greater bone marrow function for red blood cell production at 30 days of age. We suggest that sampling a small amount of blood in VLBW preterm infants can reduce the potential adverse effects of PRBC transfusion.

## Figures and Tables

**Table 1 children-09-01190-t001:** Patient characteristics between the two groups.

Characteristic	Small Volume Blood Sampling	Standard Blood Sampling	*p*-Value
	(*n* = 84)	(*n* = 20)	
	Mean ± SD	Mean ± SD	
Infant at birth			
Gestational age (week)	29 ± 2	29 ± 3	0.659
Birth weight (g)	1191 ± 245	1118 ± 302	0.463
Small for gestational age, *n* (%)	20 (23.8)	3 (15)	0.394
Male, *n* (%)	37 (44.1)	13 (65)	0.092
Multiple births, *n* (%)	32 (38.1)	6 (30)	0.499
Apgar score at 1 min	5.4 ± 1.7	5.3 ± 1.8	0.735
Apgar score at 5 min	7.0 ± 1.8	7.0 ± 1.1	0.635
Data at birth			
Hemoglobin (g/dL)	16.2 ± 1.9	18.3 ± 7.0	0.867
Hematocrit (%)	48.4 ± 5.6	45.0 ± 12.5	0.575

**Table 2 children-09-01190-t002:** Physiology measurements at day 30 in both groups.

	Small Volume Blood Sampling	Standard Blood Sampling	*p*-Value
	(*n* = 84)	(*n* = 20)	
	Mean ± SD	Mean ± SD	
Age at 1st PRBC˙ transfusion (day-old)	16.0 ± 10.4	10.8 ± 7.9	0.061
Blood loss volume via phlebotomy (mL) by day 30	22.1 ± 8.4	52.5 ± 18.4	<0.0001
PRBC˙ transfusion volume (mL) by day 30	12.8 ± 16.0	26.0 ± 16.0	0.001
Lab data at 30-day-old			
Hematocrit (%)	29.8 ± 4.0	21.4 ± 13.2	0.014
Reticulocyte (%)	5.2 ± 4.4	3.3 ± 3.6	0.008
Iron (ug/dL)	71.7 ± 20.6	48.3 ± 33.2	0.003
Ferritin (ug/L)	123.3 ± 107.5	160.9 ± 174.5	0.670
Lactate level (mmol/L)	2.0 ± 1.1	1.2 ± 0.9	0.004

PRBC˙: packed red blood cell.

**Table 3 children-09-01190-t003:** Nutrition supply.

	Small Volume Blood Sampling	Standard Blood Sampling	*p*-Value
	(*n* = 84)	(*n* = 20)	
	Mean ± SD	Mean ± SD	
Feeding amount (mL) by day 30			
Total feeding volume	4269.4 ± 2199.7	1451.6 ± 1477.7	<0.0001
Human milk/Donor human milk	2683.6 ± 1795.3	1160.2 ± 1189.4	0.001
Preterm formula	1581.8 ± 1661.2	291.5 ± 623.9	<0.0001
Duration of parental nutrition used (Day)			
	20.6 ± 20.4	48.8 ± 32.5	0.0002

**Table 4 children-09-01190-t004:** Multiple regression analysis of factors for serum iron levels.

	Regression Coefficient	95% CI *		*p*-Value
		Lower	Upper	
Feeding volume by day 30	0.002	0.002	0.010	0.002
PRBC transfusion volume by day 30	0.223	−0.107	0.777	0.115
Birth body weight	0.014	−0.045	0.479	0.254
Gestational age	1.593	−3.027	3.299	0.932
Blood loss volume via phlebotomy by day 30	0.234	−0.449	0.479	0.950

* CI: confidence interval.

**Table 5 children-09-01190-t005:** Clinical outcomes.

Outcomes	Small Blood Sampling	Standard Blood Sampling	*p*-Value
	(*n* = 84)	(*n* = 20)	
	*N* (%)	*N* (%)	
Proportion of associated condition (%)			
Respiratory distress syndrome with the need for surfactant therapy	22 (26.2)	9 (45)	0.098
Intraventricular hemorrhage	24 (28.6)	6 (30.0)	0.899
Severe intraventricular hemorrhage	2 (2.38)	2 (10.0)	0.111
Patent ductus arteriosus	44 (52.4)	10 (50.0)	0.848
Apnea of prematurity	73 (86.9)	12 (60.0)	0.093
Retinopathy of prematurity	25 (29.8)	5 (25.0)	0.749
Bronchopulmonary dysplasia	35 (41.7)	4 (20.0)	0.072
NEC ^1^, all stage	5 (5.95)	1 (5.0)	0.869
Confirmed NEC ^1^ (≥2a Bell’s stage)	1 (1.2)	1 (5.0)	0.265
Culture proved sepsis	10 (11.9)	9 (45.0)	0.001
Duration of associated condition (days) (Mean ± SD)			
Time to regain birth body weight	12.7 ± 5.0	18.1 ± 7.0	0.003
Time on a ventilator or nasal CPAP ^2^	43.5 ± 23.6	32.3 ± 25.2	0.073
Time on supplemental oxygen	1.3 ± 3.7	1.1 ± 2.8	0.966
Hospital stays	67.2 ± 23.7	84.2 ± 44.2	0.181

NEC ^1^: necrotizing enterocolitis; CPAP ^2^: nasal continuous positive airway pressure.

**Table 6 children-09-01190-t006:** Multiple regression analysis of factors for complications related to oxidative stress.

			PRBC Transfusion Volume by Day 30	Birth Body Weight	Blood Loss Volume via Phlebotomy by Day 30	Gestational Age
IVH ^1^	Regression coefficient		0.021	0.001	0.018	0.164
	95% CI *	Upper	0.080	0.005	0.024	0.241
		Lower	−0.003	−0.0003	−0.046	−0.402
	*p*-value		0.067	0.096	0.537	0.624
NEC ^2^	Regression coefficient		0.042	0.002	0.031	0.312
	95% CI *	Upper	0.180	0.007	0.042	0.800
		Lower	0.017	−0.002	−0.080	−0.424
	*p*-value		0.018	0.296	0.536	0.548
BPD ^3^	Regression coefficient		0.020	0.001	0.018	0.146
	95% CI *	Upper	0.079	0.001	0.010	0.348
		Lower	−0.002	−0.003	−0.061	−0.227
	*p*-value		0.064	0.137	0.165	0.680
ROP ^4^	Regression coefficient		0.022	0.001	0.020	0.187
	95% CI *	Upper	0.066	0.002	0.022	0.051
		Lower	−0.021	−0.003	−0.055	−0.690
	*p*-value		0.315	0.600	0.429	0.117

* CI: confidence interval; IVH ^1^: intraventricular hemorrhage; NEC ^2^: necrotizing enterocolitis; BPD ^3^: bronchopulmonary dysplasia; ROP ^4^: retinopathy of prematurity.

## Data Availability

The data presented in this study are available on request from the corresponding author.

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
