# Peer review of "Effect of Small Volume Blood Sampling on the Outcomes of Very Low Birth Weight Preterm Infants"

_children, 2022, doi:10.3390/children9081190_

Round 1

Reviewer 1 Report

Comments to the Author

The manuscript entitled “Effect of small volume blood sampling on the outcomes of very low birth weight preterm infants" focuses on an interesting topic in perinatal medicine. Particularly, authors attempted to evaluate the effects of blood loss reduction due to phlebotomy on anemia of prematurity (AOP), packed red blood cell (PRBC) transfusion, and outcome in VLBW preterm infants.

The aims of the study are within the scope of the journal; however, the manuscript has some major flaws.

Method section

  • Material and Method section needs to better guide and clarify the design of the study.
  • Redundant information has to be skipped from M & M section (i.e. on page 2, lines 62-63 the Authors reported the same information described on page 2, lines 50-51).

Statistical analysis

 ·         Statistical analysis needs to be expanded providing sample size calculation.

·         Statistical section should be reported in a different paragraph.

Result section

·        Results section is difficult to follow. I suggest Authors to re-write organizing data in separate paragraph avoiding redundant information (i.e. please reported the results either in tables or main text, i.e page 3 lines 118 – table 2)

 Discussion section

  • The discussion section is too long and difficult to read. It should be focused on the main aim of the study. It is necessary to revised this section, discussing more in depth the results obtained.

References

·         The bibliography needs to be updated.

Please avoid typing errors in all manuscript sections.

 It would be better if you would kindly revise the manuscript by a native English

 speaker.

Reviewer 2 Report

In this retrospective study, the effects of a micromethod blood extraction policy in low birth weight premature infants are considered.

In the historical cohort or control group, the blood volumes extracted were 3 mL per sample. In the new method cohort, 0.2 mL of blood is drawn from each sample.

Due to the immaturity of prematurity, this vulnerable population is susceptible to anemia, in addition to other factors such as the half-life of erythrocytes, possible infections, jaundice due to group incompatibility that leads to hemolysis, and nutritional deficiencies.

Blood draws are a major cause of anemia in preterm infants. 3ml per sample is too much blood for a blood count or blood gas.

The best conditions at birth also influence that they are more stable newborns and that they need less extractions.

It is missing to know the state of previous maturation with prenatal corticosteroids, as well as the presence of intrauterine growth retardation in the sample.

The delay in clamping the umbilical cord also influences the hematocrit and stability at birth.

Regarding the lactate elevation, we do not know the hemodynamic status of the newborns at the time it was determined.

Having a more aggressive feeding policy that allows parenteral nutrition and venous access to be withdrawn early may probably have an influence on the lower incidence of sepsis in the operated group.

Perhaps due to the sample size, no differences were found in the morbidities of prematurity.

Despite a higher hematocrit and better oxygen transport, the newborns in the exposed group had higher rates of bronchopulmonary dysplasia and longer mean time on respiratory support, although not significantly.

It is reasonable to conclude that if less blood volume is lost in each blood draw, the preterm reserve will be greater.

perhaps they are populations with less favorable perinatal conditions.

Reviewer 3 Report

The authors published a paper in 2009 investigating the effect of blood transfusions on the outcome of very low birthweight (VLBW) preterm infants under different transfusion criteria. Based on the results they modified their policy regarding indication for transfusion (restrictive criteria).

In 2016 their NICU implemented small volume blood sampling.

In the present retrospective study Pin-Chun Su et al compared the clinical data of VBLW infants using small volume blood sampling (study group, n=84) to VLBW infants undergone standard blood sampling (control group, data from 2009, n=20). Both groups received transfusion under restrictive criteria. Blood loss due to phlebotomy and transfusion volume examined in 30-day-old infants were significantly lower in the study group than in the control group.

Hematocrit, reticulocyte, and iron levels were significantly higher in the study group compared to controls. The incidence of patent ductus arteriosus, severe intraventricular hemorrhage, retinopathy of prematurity, and bronchopulmonary dysplasia were similar.

The small volume blood sampling resulted in lower transfusion volume, less severe anemia, and better bone marrow function.

To this retrospective study premature infants with gestational age under 32 weeks, and birth weight less than 1500 g were enrolled who were admitted to the NICU between June 2016 and October 2018 as the study group (sampling small amount of blood).

Infants from their previous study (published in 2009) were used as controls.

At that point any reviewer should say: I stop evaluating, the scientific value is questionable. To compare data obtained 10 years apart is not acceptable.

However; it is obvious that withdrawing 10 times less blood volume will result in higher hematocrit value, less need for transfusions and better clinical outcome.

Professional English editing is required.

The list of references does not include important recent publications, for example:

https://pubmed.ncbi.nlm.nih.gov/33382931/

https://pubmed.ncbi.nlm.nih.gov/18653585/

https://pubmed.ncbi.nlm.nih.gov/33472464/

This paper is not suitable for publication.

Round 2

Reviewer 1 Report

I have carefully reviewed the revised version of the manuscript entitled “Effect of small volume blood sampling on the outcomes of very low birth weight preterm infants

I would like to thank the authors; they have addressed my comments raised on their original submission. However, I still have some observations.

Materials and Methods section: please rename the paragraph 2.2 (see paragraph 2,1)

Result section needs further revision. Please provide separate chapters.

The tables are still difficult to read. I would suggest for example to skip mean/sd form the main part of the tables.

Author Response

Response to Reviewer 1

Point 1:

Materials and Methods section: please rename the paragraph 2.2 (see paragraph 2,1)

Response 1:

Thank you for your comment. We have revised the title of paragraph 2.2 as Ethics statement

Point 2:

Result section needs further revision. Please provide separate chapters.

Response 2:

Thank you for your comment. We have added subtitles in the Result section.

Point 3:

The tables are still difficult to read. I would suggest for example to skip mean/sd form the main part of the tables.

Response 3:

Thank you for your comment. Because we would like to present the details of our data, we do not skip mean/sd in the Tables. It would be better to read after we added subtitles for the Result section.

Reviewer 2 Report

Thank you for your responses to my suggestions about your manuscript.

the score on the apgar test undoubtedly helps to interpret the state of the newborn at birth, but, in my opinion, the gestational pathology is important to know if the cohorts are comparable to each other, as well as the maturation with prenatal corticosteroids. A more stable newborn needs fewer analytical determinations and, therefore, will have less risk of iatrogenic anemia.

a section on strengths and weaknesses of his research is missing.

I have not found the calculation of the sample size necessary to find differences between both cohorts.

Author Response

Response to Reviewer 2

Point1:

the score on the apgar test undoubtedly helps to interpret the state of the newborn at birth, but, in my opinion, the gestational pathology is important to know if the cohorts are comparable to each other, as well as the maturation with prenatal corticosteroids. A more stable newborn needs fewer analytical determinations and, therefore, will have less risk of iatrogenic anemia.

Response 1:

Thank you for your comment. We really appreciate your suggestions. However, prenatal corticosteroid status was inconvincible due to 45% missing data in the control group (shown in the following table), which is why it was not included in this study. Otherwise, no significant differences were observed in the proportion of infants with respiratory distress syndrome (RDS) with the need for surfactant therapy between the two groups (We have added these data to Table 6). Respiratory condition at birth was considered comparable in terms of the need for analytical blood determinations thereafter.

Control group

Study group

P value

Prenatal steroid, N (%)

6 (30)

64 (76.2)

< 0.001

(By Chi square test)

No prenatal steroid, N (%)

5 (25)

20 (23.8)

Unknown status for prenatal steroid, N (%)

9 (45)

0 (0)

Point 2: 

a section on strengths and weaknesses of his research is missing.

Response 2:

Thank you for your suggestions. We have added a paragraph for the strengths and weaknesses of this study (Lines 254-266).

Point 3:

I have not found the calculation of the sample size necessary to find differences between both cohorts.

Response 3:

We tried to calculate the power based on the current sample size of our study by using JMP 14.2.0 desktop statistical discovery software (SAS Institute Inc., USA) (as shown in the following figure--Please see the attachment), and the result was 97.84%, indicating that the sample size of our study was sufficiently large.

Reviewer 3 Report

The quality of this paper is improved, but my biggest problem is still related to the control group. The authors claimed :

"We appreciate your advice. We agree that comparing the data obtained ten years apart is not acceptable."

All of my other suggestions were taken into consideration, they revised the list of references also. 

Author Response

Response to Reviewer 3

Point:

The quality of this paper is improved, but my biggest problem is still related to the control group.

Response:

Thank you for your comment. We have added an explanation to the limitation paragraph describing the challenge with the data from the control group in this study. (Lines 254-266)

Round 3

Reviewer 2 Report

thanks for your modifications

Author Response

Thank you for your comments and suggestions to make our manuscript better.

Reviewer 3 Report

You added a couple of sentences to explain the reason why the "historical" control group was used.

I respect your ethical concern but there is a scientific issue and I am still not fully convinced.

Author Response

Point: You added a couple of sentences to explain the reason why the "historical" control group was used. I respect your ethical concern but there is a scientific issue and I am still not fully convinced.

Response:

Thank you for your comment. We really appreciate your suggestions. We think your concern for the historical group is about the inequality between these two groups. The condition after birth was determined by the APGAR scores. In our study, we showed no statistical differences in the first minute and 5th minute APGAR scores of the study group and the control group (Table 1). Besides, no significant differences were observed in the proportion of infants with respiratory distress syndrome (RDS) with the need for surfactant therapy between the two groups (Table 6). Therefore, respiratory condition at birth was comparable to the need for analytical blood determinations.

We have added the description above into the paragraph for the strengths and weaknesses of this study (Lines 260-266).

Thank you again for your comment.